# Internalizing and Externalizing Behaviors: A Cross-Cultural Study in Colombian and Mexican Adolescents with Eating Disorders

**DOI:** 10.3390/ijerph22060932

**Published:** 2025-06-13

**Authors:** Jaime Humberto Moreno Méndez, María Margarita Rozo Sánchez, Natalia Maldonado Avendaño, Andrés Mauricio Santacoloma Suárez, Julieta Vélez Belmonte, Jesús Adrián Figueroa Hernández, Stephanie Tanus Minutti, Rodrigo César León Hernández

**Affiliations:** 1Psychology Faculty, Catholic University of Colombia, Bogotá 111311, Colombia; jhmoreno@ucatolica.edu.co (J.H.M.M.); mmrozos@ucatolica.edu.co (M.M.R.S.); nmaldonadoa@ucatolica.edu.co (N.M.A.); amsantacoloma@ucatolica.edu.co (A.M.S.S.); 2School of Psychology, Anáhuac University, Puebla 72810, Mexico; julieta.velez@anahuac.mx (J.V.B.); adrian.figueroa@anahuac.mx (J.A.F.H.); stephanie.tanus@anahuac.mx (S.T.M.); 3Secretariat of Science, Humanities, Technology and Innovation, Center for Psychological Specialties NEANDI, Mexico City 03810, Mexico

**Keywords:** adolescents, cross-cultural study, eating disorders, emotional problems, externalizing behaviors, internalizing behaviors

## Abstract

In Colombia and Mexico, an increase in emotional, behavioral, and eating problems in adolescents has been documented after the pandemic. The objective was to characterize the relationship between internalizing and externalizing behaviors in adolescents with eating disorders in Colombia and Mexico according to the adolescents’ self-report and the parents’ report. In Colombia, 17 adolescents between 12 and 18 years old (*M* = 15.4; *SD* = 1.8) and one of their parents (*n* = 17); in Mexico, 8 adolescents between 12 and 17 years old (*M* = 14.6; *SD* = 1.6) and one of their parents (*n* = 8) were evaluated. The parents completed the Child Behavior Checklist (CBCL), and the adolescents completed the self-report (YSR) and the EAT-26. The analyses showed a statistically significant correlation between eating problems and anxiety/depression of the YSR (*r* = 0.39; *p* = 0.031). Statistically significant differences (*p* < 0.05) were found in the CBCL scores for externalizing problems, somatic complaints, and rule-breaking behavior; all scores were higher in the Colombian sample. The findings provided partial support for differences between adolescents with eating disorders and parental reports. A higher percentage of clinical levels was reported by adolescents compared to their parents, except for the anxious/depressive and aggressive behavior subscales.

## 1. Introduction

Eating, emotional, and behavioral disorders are among the mental health issues that became most evident following the COVID-19 pandemic. In this sense, several studies have reported a prevalence of emotional symptoms, which range between 22% and 24.9% for anxiety and 19.7% to 29% for depression [1,2,3,4]. In another study, behavior problems were reported in 14.34% of cases for antisocial behavior and 5% for criminal behavior [5]. A systematic review carried out on 42 studies worldwide found that in 17% of the research analyzed, fears and stress derived from the pandemic increased eating disorders. These problems stem from factors such as the loss and health impacts on family and friends, an increase in domestic violence and child abuse, financial difficulties experienced by parents, and social distancing, among others (Pan American Health Organization) [6].

One of the most empirically supported perspectives on emotional and behavioral problems in adolescents is the Achenbach System of Empirically Based Assessment (ASEBA). In this system, it has been possible to determine, through multivariate analysis, syndromes that group clinical problems that co-occur in a dimensional manner [7]. This system classifies anxious/depressive symptoms, withdrawal/depressive symptoms, and somatic complaints under the broad category of broadband internalizing problems that are expressed as overcontrolled patterns of emotional and affective behavior, while aggressive and rule-breaking behaviors are grouped under externalizing broadband problems, which are underregulated disruptive behavior patterns [8].

In relation to these psychological problems in adolescents, underlying transdiagnostic processes have been found, such as negative affectivity, impulsivity, cognitive biases, and difficulties in emotional regulation, which make them more likely to show risky behaviors for their mental health [9].

Empirical evidence indicates that adolescents with internalizing and externalizing problems frequently exhibit eating disorders (EDs) [10,11]. EDs are characterized by concerns about weight, height, body shape, and image, as well as disrupted eating habits [12].

In a scoping review conducted in Colombia with adolescents, a correlation was found between eating disorders of anorexia and bulimia and internalizing symptoms of an anxious-depressive type [13]. In a comparative study conducted in Mexico, it was found that the risk of developing bulimia and anorexia is higher when anxiety is high [14]. These studies have suggested further research in this field, given that no clear relationship between eating disorders and externalizing problems has been demonstrated.

Given the relatively recent emergence of this issue worldwide, there is increasing interest in investigating the relationship between internalizing and externalizing problems and EDs in adolescent populations [15,16,17,18,19]. However, further studies with clinical samples are needed.

Additionally, adolescents of both genders with externalizing problems are more likely to engage in binge drinking [15,16,17]. It has also been documented that adolescents with EDs report fewer externalizing behaviors, such as rule-breaking, compared to what their parents report. This discrepancy highlights the need for a more comprehensive assessment based on reports from both parents and adolescents [10].

In Latin America, some studies have explored the relationship between internalizing and externalizing problems and EDs. However, there is no empirical evidence that explores the differences between clinical samples from these Latin American countries. In Colombia, partial data have been reported on the prevalence of eating problems in studies conducted before the pandemic with non-clinical samples. For example, a study in Bogotá involving 937 adolescents found that 53.7% had eating problems; furthermore, the psychological problems associated with EDs included a 23.3% prevalence of depression and a 10% prevalence of anxiety [20]. Another study by Fajardo et al. [21] reported a 30.1% prevalence of potential EDs, with women being at greater risk (41.3%).

In another study conducted in Colombia with 40 patients aged 11 to 19 years with EDs, 72.5% exhibited internalizing behaviors, 65% showed anxious symptoms, 30% showed depressive symptoms, 42.5% exhibited externalizing behaviors, and 22.5% showed aggression [22]. However, this study only considered the adolescents’ perspective, excluding other informants such as parents, whose inclusion would be important in future research.

In Mexico, a study involving 3144 adolescents aged 13 to 17 reported that 5.6% met the criteria for potential EDs, with a higher prevalence in females (9.6%) compared to males (1.8%) [23]. Moreover, adolescents insecure about their bodies tend to internalize an aesthetic model that values thinness, increasing their risk of EDs by 11.8 times [24,25].

A subsequent study by Villalobos Hernández et al. [26] reported a higher risk of EDs among female adolescents (2.0%) compared to males (1.2%). The study concluded that further research is essential to provide a broader perspective on the variables associated with EDs in Mexican adolescents.

Considering the above, it is crucial to explain the psychological mechanisms underlying the association between internalizing and externalizing symptoms and EDs in adolescence. A study by Zancu and Diaconu-Gherasim [27] concluded that perceived weight stigma is positively associated with internalizing symptoms and weight bias. Additionally, learned beliefs about weight were negatively related to body esteem and positively associated with internalizing symptoms and EDs.

In this context, Moreno Encinas et al. [28] indicated that internalization is a general risk factor for the psychopathology of EDs and depressive disorders. EDs are also associated with an obsession with thinness, inefficacy, interoceptive awareness, depression, trait anxiety, and obsessive-compulsive symptoms. Furthermore, limited parental and peer connections are associated with externalizing and internalizing problems and EDs in adolescents [28,29]. Parents’ difficulties in identifying their children’s emotional and eating problems have also been documented, which is reflected in a discrepancy in the reporting of these problems between adolescents and their parents, limiting the seeking of timely treatment for their children [30].

The tripartite model of EDs posits that family, friends, and the media exert a dysfunctional influence on body image and internalizing symptoms in adolescents with EDs [31]. However, a study conducted in Mexico found that family pressure had a more significant impact than peer and media pressure on the body image of adolescents with EDs through negative messages, eating behaviors, and home-based diets. These factors may be associated with the fact that in Latin American contexts, families, especially mothers, tend to have a significant influence on their children’s eating behaviors [32].

In the Colombian cultural context, a relationship has been documented between a history of eating disorders in parents, criticism and jokes about their children’s body shape and weight, and eating problems in adolescents, which ends up leading them to associate success and social approval with thinness, especially in women [20]. Adolescents with EDs are among the most vulnerable groups for mental health problems, given their heightened risks of emotional and behavioral issues. Paradoxically, they are one of the least studied groups, particularly in the Latin American context.

Furthermore, there has been a lack of more studies with clinical samples, both in Colombia and Mexico, that account for the relationships between internalizing and externalizing problems and EDs from the perspective of adolescents and their parents, especially if one takes into account that some of the findings presented have reported variability on the association between these variables in samples of school-aged adolescents [20,21,22,26].

Therefore, the objective of this study was to establish the relationship between internalizing and externalizing behaviors in adolescents with eating disorders in Colombia and Mexico, based on adolescents’ self-reports and parental reports. To address this objective, the following hypotheses were proposed: (H1) There are statistically significant differences in internalizing and externalizing behaviors among adolescents with EDs in Colombia and Mexico. (H2) EDs are more strongly associated with internalizing behaviors than externalizing behaviors. (H3) There are differences in the reports of Colombian and Mexican parents and adolescents regarding internalizing, externalizing, and eating problems (Figure 1).

## 2. Materials and Methods

This study is a cross-sectional observational study with an associative strategy to conduct a comparative and predictive study. The differences between a Colombian and a Mexican sample were examined, exploring the relationships between the various variables analyzed [33]. (internalizing behaviors, externalizing behaviors, and eating problems).

### 2.1. Participants

An incidental convenience sampling method was used, which refers to the process of selecting a representative sample that meets the necessary characteristics for the research [34]. Inclusion criteria required adolescents to be between 12 and 18 years old, live with at least one parent, and consent to participate in the study through parental informed consent and adolescent assent. Participants were also required to be receiving treatment for an eating disorder such as anorexia, bulimia, or binge eating at private clinics specializing in these conditions, one in Bogotá and one in Mexico City.

Treatment duration for the Mexican sample ranged from 4 months to 3 years, while in the Colombian sample, it ranged from 1 month to 3 years. The main exclusion criterion was the presence of neurodevelopmental disorders in the adolescents.

### 2.2. Instruments

Sociodemographic Data: These data include collected information on the adolescents’ age, gender, and educational level, as well as the age and gender of their parents.

Child Behavior Checklist (CBCL) Parent Format by Achenbach and Rescorla (2001) [8]: This instrument comprises 112 items designed to identify behavioral syndromes in clinical samples. Responses are scored as follows: (0 = not true, 1 = somewhat true, 2 = very true/often true). Raw scores are converted into T-scores categorized into ranges: Normal (50–64), Borderline (65–69), and Clinical (70–100). The interclass correlation coefficient (ICC) reliability was 0.952, with a reliability of 0.83 and internal consistency of 0.94 in the Colombian population [33,34]. Validity evidence is available for both Colombian [35] and Mexican samples [36].

Youth Self-Report (YSR) Adolescent Format by Achenbach and Edelbrock (2001) [8]: This self-report form contains 112 items identifying behavioral syndromes in clinical samples of adolescents aged 11–18 years. Items are scored on a 3-point Likert scale (0 = not true, 1 = somewhat true, 2 = very true/often true). Raw scores are converted into T-scores categorized as Normal (50–64), Borderline (65–69), and Clinical (70–100). The Pearson correlation for ages 11–14 was 0.77, and for ages 15–18 it was 0.89. This scale has been previously applied in Colombian [37] and Mexican samples [38,39].

Eating Attitudes Test-26 (EAT-26) by Garner and Garfinkel (1979) [40]: This test examines adolescent symptoms and concerns related to eating disorders. Validity evidence exists for Colombian [41] and Mexican populations [42]. Factor analysis revealed four subscales: Bulimia: 6 items (e.g., “I feel the urge to vomit after meals”); Dieting: 6 items (e.g., “I eat diet foods”); Food Preoccupation: 7 items (e.g., “I avoid eating when I am hungry”); and Oral Control: 5 items (e.g., “I cut my food into small pieces”).

Responses are scored on a 6-point scale (never, rarely = 0; often = 1; very often = 2; always = 3). Higher total scores indicate a greater risk of anorexia or bulimia, with a cutoff score of ≥11. Reliability was 93.6%.

### 2.3. Procedure

In the first phase, the study was approved by the University Ethics Committee that led the study, according to Minutes No. 33 of 17 October 2023. Institutions treating ED populations in Colombia and Mexico were contacted to explain the study and obtain approval for participant recruitment. In the second phase, the sample was selected after explaining the project and obtaining parental informed consent and adolescent assent. The researchers created an online version of the EAT-26 questionnaire and the YSR for adolescents and administered the CBCL to parents. These instruments were administered in person at the clinics where adolescents received treatment. Data analysis was conducted in the final phase.

### 2.4. Data Analysis Plan

Descriptive statistics, including mean, median, standard deviation, skewness, and kurtosis, were calculated for quantitative variables, while frequencies and percentages were determined for categorical variables. The Shapiro–Wilk test was used to verify the normality of quantitative variables.

Spearman’s correlation was employed to analyze the relationship between CBCL, YSR, and EAT-26 scores when data did not follow a normal distribution; Pearson’s correlation was used otherwise. The Mann–Whitney U test was applied for non-normally distributed data when comparing nationalities, and Welch’s *t*-test was used for normally distributed data due to variance differences detected with Levene’s test. To compare EAT-26 scores across risk levels identified by the CBCL and YSR while considering the effect of nationality, a two-way fixed-factor ANOVA was used. All analyses were performed using Jamovi software (Version 2.3, https://www.jamovi.org, accessed on 15 March 2025) [43].

## 3. Results

The objective of this study was to characterize the relationship between internalizing and externalizing behaviors in adolescents with EDs from Mexico and Colombia, based on adolescents’ self-reports and parental reports. A total of 25 adolescents (2 males and 23 females) aged 12 to 18 years were evaluated (Males: *M* = 13, *SD* = 1.41; Females: *M* = 15.3, *SD* = 1.66). The average age of the Colombian sample was 15.4 years (*SD* = 1.8), and of the Mexican sample, 14.1 (*SD* = 2.15). Table 1 shows the description of the sociodemographic characteristics according to the country and the estimation of differences for each country based on sociodemographic variables.

When comparing the variables of interest according to sociodemographic characteristics in each country, it was observed that, in the Colombian sample, there were statistically significant differences between sexes on the YSR internalizing problems variable, with higher scores among women (*M* = 27.87; *SD* = 10.52) than among men (*M* = 19; *SD* = 2.83). Similarly, on the YSR aggressive behavior variable, women also obtained higher scores (*M* = 12.67; *SD* = 4.03) compared to men (*M* = 5; *SD* = 2.83).

In the Mexican sample, statistically significant differences were found according to the parents’ occupation in the YSR internalizing problems variable between the employee (*M* = 29.75; *SD* = 4.99) and independent workers (*M* = 16; *SD* = 1.41) levels, as well as between employees and homemakers (*M* = 19.33; *SD* = 2.52); however, no significant differences were observed between independent workers and homemakers.

In response to hypothesis 1, the corresponding findings are presented in Table 2. Descriptive analyses were conducted for the total sample and separately by nationality. The results showed that internalizing and externalizing problems reported by parents and adolescents were higher in the Colombian sample, as was the mean score on the EAT-26. In the total sample, the data distribution was not normal for three of the CBCL subscales (Withdrawn/Depressed, Somatic Complaints, and Rule-Breaking Behavior) (*p* < 0.05). Similarly, in the YSR subscales, a normal distribution was not found for Rule-Breaking Behavior and Aggressive Behavior scores (Table 2).

In response to hypothesis 2, correlation analyses between the CBCL, YSR, and EAT-26 scales identified a statistically significant correlation between the EAT-26 score and the YSR Internalizing Anxious/Depressed subscale (*r* = 0.39; *p* = 0.031) (Table 2).

Additionally, comparisons between nationalities were conducted for the CBCL and YSR subscales and the total EAT-26 score (Table 2). Statistically significant differences (*p* < 0.05) were found in the CBCL scores for externalizing problems, somatic complaints, and rule-breaking behavior, with effect sizes suggesting clinical relevance [36]; all scores were higher in the Colombian sample. However, no statistically significant differences were observed between countries for the EAT-26 score or the YSR subscales (*p* > 0.05).

However, when comparing risk levels for internalizing and externalizing problems, a higher percentage is observed in the clinical levels of the YSR compared to the CBCL, except for the Withdrawn/Depressed and Aggressive Behavior subscales. Additionally, across all CBCL and YSR subscales, clinical-level scores show a higher percentage in the Colombian sample than in the Mexican sample (Table 3).

In the Rule-Breaking Behavior subscale of the CBCL, all cases in the Mexican sample fall within the no-risk category, differing from the situation observed in the Colombian sample. Regarding the YSR subscales, it is notable that in the Internalizing Problems subscale, neither sample has individuals classified at risk, with most scoring in the clinical range. In contrast, in the Aggressive Behavior subscale, participants are primarily classified as no-risk or at-risk, while the percentage of clinical scores is low (Table 3).

Finally, in response to Hypothesis 3, EAT-26 scores were compared between the risk levels identified by parents on the CBCL and those reported by adolescents on the YSR, including the effect of nationality in the model. For the CBCL subscales, no statistically significant differences were found (*p* > 0.05), and the same was true for most of the YSR subscales. However, in the YSR Anxious/Depressed subscale, an interaction effect between risk levels and nationality was observed, with an effect size of sufficient clinical relevance [44].

This model suggests that risk assessment scores do not follow a uniform trend that can be explained solely by risk level or nationality. In other words, the relationship between risk level (no risk, at risk, clinical) and scale scores varies depending on nationality (Colombian vs. Mexican) (Table 4). These findings could indicate that Colombian adolescents with eating disorders tend to report more anxious-depressive symptoms compared to Mexican adolescents.

## 4. Discussion

The results of the present research showed that internalizing and externalizing problems reported by parents and adolescents were higher in the Colombian sample, as was the mean score on eating problems, an aspect that could be explained by differences in the internalization of culturally imposed body image standards [40]. It could also be hypothesized that there was greater recognition of problems in Colombian adolescents compared to Mexican adolescents, although discrepancies in sample size could also have influenced these results, given that the number of participants evaluated in Colombia was more than double that of Mexico. These findings show partial support for hypothesis 1 and show the recognition of these problems by adolescents, as has been shown in other studies with Latin American adolescents [11,13,14,20,21]. These results are possibly linked to a greater emotional sensitivity of these adolescents to criticism about their body image in the family environment and to a need for social approval [20]. Furthermore, these results have important clinical implications, since a fundamental aspect of accessing mental health care treatment is first recognizing that one has a psychological problem.

Correlation analyses between the CBCL, YSR, and EAT-26 scales identified a statistically significant relationship between eating problems and anxious/depressive symptoms, validating hypothesis 2, which posited a stronger association between eating disorders and internalizing behaviors. It is important to note that most participants scored in the clinical range for anxious/depressive symptoms, internalizing problems, and eating problems. In contrast, no significant correlation was found between externalizing behavior indicators and eating problems. In this regard, an association has been found between a negative evaluation of body weight and anxious and depressive symptoms, which leads to negative attitudes and unregulated eating behaviors in adolescents [45,46].

These findings are consistent with those of Moreno Encinas et al. [23] and suggest that internalizing problems in adolescence may be a factor associated with eating disorders. Likewise, it could be explained because adolescents with EDs and the familiar and social media exerted a dysfunctional influence on body image and internalizing symptoms in adolescents with EDs [13,20,31]. This insight can help identify high-risk groups and contribute to better prevention strategies or targeted treatment objectives at an early stage of the disorder. Prevention programs were found to be more effective when samples were divided into high- and low-risk groups for eating disorders.

Regarding hypothesis 3, which examined differences in reports from parents and adolescents in Colombia and Mexico on internalizing, externalizing, and eating problems, the findings provided partial support for differences between adolescent and parental reports. A higher percentage of clinical levels was reported by adolescents compared to their parents, except for the anxious/depressive and aggressive behavior subscales. This discrepancy aligns with the findings of Mensi et al. [10] and Crowther et al. [30], who also reported differences in the perception of these problems.

These results emphasize the need for early detection using a multicomponent evaluation that includes input from parents, educators, and peers. However, particular attention should be given to adolescents’ self-assessments, as they reported a higher percentage of internalizing problems than identified by their parents in this study. This approach can help prevent psychological problems, particularly internalizing disorders, and address behavioral consequences of weight stigma and bias [47].

The findings highlight the need to improve mental health among adolescents in both countries, consistent with Chavira et al. [48], who emphasize the importance of developing evidence-based studies to address these issues. Such research can inform funding for prevention and intervention programs and propose changes to public policies to more effectively meet the mental health needs of adolescents in the post-pandemic period, especially those at higher risk, such as those with eating disorders.

It is important to note that eating disorders are associated with other psychopathological variables, and the results related to internalization may be linked to comorbidities with anxiety and depression disorders. Future research should include a review of participants’ medical histories to identify comorbidities and incorporate other diagnoses. Similarly, it is suggested that the sample size be increased and that independent analyses be conducted among the diagnostic entities of eating disorders.

The limitations of this study lie in the small and unequal sample sizes between Mexico and Colombia and the convenience sampling in both countries, making it difficult to generalize the results to either country or even the region. It is important to acknowledge that the small and unequal sample sizes by country (*n* = 17 for Colombia, *n* = 8 for Mexico) represent a limitation that may have influenced the statistical power of the study. Small sample sizes make it hard to find significant effects, especially if they are small or moderate. Additionally, group size imbalance can affect the stability of estimates and limit the generalizability of the findings. In this regard, although effect sizes were reported alongside *p*-values, they should be interpreted with caution, as effect sizes in small samples can be unstable or overestimated [49].

Qualitative data were not obtained that would allow contextualizing the discrepancies between the reports of adolescents and parents since the clinics only allowed the application of the instruments due to confidentiality issues regarding patient information.

Future studies are needed with larger and more balanced samples across countries, complementing the collection of information from adolescents with qualitative assessment strategies that contribute to greater consistency and robustness of the findings.

## 5. Conclusions

These results emphasize the need for early detection through a multicomponent assessment that includes the participation of parents and peers. However, special attention should be paid to adolescents’ self-reports, as they reported a higher percentage of internalizing problems than their parents in this study. This approach may help prevent psychological problems, particularly internalizing disorders, and address the behavioral consequences of weight-related stigma and bias.

This study provides evidence on the relationship between internalizing and externalizing behaviors in a clinical sample of adolescents from Colombia and Mexico with eating disorders and includes the perspectives of parents and their children. Therefore, it is suggested that the results of this study be interpreted with caution due to the sample size.

## Figures and Tables

**Figure 1 ijerph-22-00932-f001:**
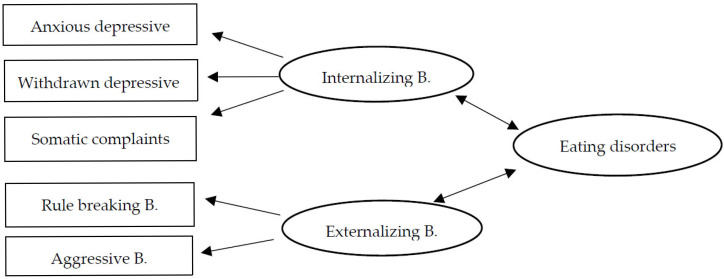
Hypothetical model of the relationship between internalizing and externalizing behaviors in Colombian and Mexican adolescents with eating disorders.

**Table 1 ijerph-22-00932-t001:** Description of the sociodemographic characteristics by country and the estimation of differences for each country based on the sociodemographic variables.

Sociodemographic Characteristics		Frequency (Percentage)	Comparison Analysis	CBCL							YSR							EAT-26
				AN-D	AI-D	QS	PI	RR	CA	PE	AN-D	AI-D	QS	PI	RR	CA	PE	
Colombian																		
Gender	Male	2 (7.7)	Statistic value	−0.656 a	13.5 b	10 b	−0.47 a	13 b	−108 a	−0.914 a	−2.66 a	−1.03 a	−0.703 a	−2.63 a	9 b	1.0 b	−2.79 a	−0.396 a
	Female	15 (57.7)	*p*	0.618	0.879	0.50	0.714	0.822	0.46	0.506	0.073	0.321	0.595	0.036	0.409	0.043	0.117	0.749
			ES	−0.546	0.10	0.333	−0.406	0.133	−0.91	−0.725	−1.347	−0.376	−0.569	−1.151	0.400	0.933	−1.675	−0.305
Parents’ Gender	Male	5 (19.2)	Statistic value	−0.989 a	22.5 b	27.5 b	−0.819 a	22 b	0.06 a	−0.408 a	−21.9 a	−14.5 a	−19.5 a	−23.1 a	19 b	25.5 b	−10 a	−0.379 a
	Female	12 (46.2)	*p*	0.34	0.451	0.832	0.431	0.425	0.953	0.691	0.056	0.203	0.082	0.055	0.265	0.672	0.344	0.713
			ES	−0.4667	0.250	0.083	−0.409	0.267	0.029	−0.203	−11.21	−0.83	−0.998	−12.59	0.367	0.150	−0.517	−0.1903
Grade	Sixth	1 (3.8)	Statistic value	NE	NE	NE	NE	NE	NE	NE	NE	NE	NE	NE	NE	NE	NE	NE
	Seventh	2 (7.7)																
	Eighth	0 (0)	*p*	NE	NE	NE	NE	NE	NE	NE	NE	NE	NE	NE	NE	NE	NE	NE
	Ninth	5 (19.2)																
	Tenth	3 (11.5)	ES	NE	NE	NE	NE	NE	NE	NE	NE	NE	NE	NE	NE	NE	NE	NE
	Eleventh	6 (23.1)																
	Twelfth	0 (0)																
Parents’ Occupation	Self-employed	4 (15.4)	Statistic value	NE	NE	NE	NE	NE	NE	NE	NE	NE	NE	NE	NE	NE	NE	NE
	Employee	12 (46.2)	*p*	NE	NE	NE	NE	NE	NE	NE	NE	NE	NE	NE	NE	NE	NE	NE
	Homemaker	0 (0)	ES	NE	NE	NE	NE	NE	NE	NE	NE	NE	NE	NE	NE	NE	NE	NE
	Other	1 (3.8)																
Mexican																		
Gender	Male	0 (0)	Statistic value	NE	NE	NE	NE	NE	NE	NE	NE	NE	NE	NE	NE	NE	NE	NE
	Female	9 (34.6)	*p*	NE	NE	NE	NE	NE	NE	NE	NE	NE	NE	NE	NE	NE	NE	NE
			ES	NE	NE	NE	NE	NE	NE	NE	NE	NE	NE	NE	NE	NE	NE	NE
Parents’ Gender	Male	2 (7.7)	Statistic value	−1.43 a	5.50 b	4.50 b	−1.35 a	2.50 b	1.237 a	1.618 a	−0.219 a	−2.27 a	0.826 a	0.107 a	7 b	5 b	−0.883 a	4.17 a
	Female	7 (26.9)	*p*	0.232	0.767	0.551	0.295	0.219	0.262	0.151	0.854	0.058	0.551	0.930	1	0.659	0.435	0.112
			ES	−0.945	0.214	0.357	−0.989	0.643	0.661	0.930	−0.187	−1.281	0.787	0.099	0	0.286	−0.588	0.876
Grade	Sixth	1 (3.8)	Statistic value	NE	NE	NE	NE	NE	NE	NE	NE	NE	NE	NE	NE	NE	NE	NE
	Seventh	1 (3.8)																
	Eighth	3 (11.5)	*p*	NE	NE	NE	NE	NE	NE	NE	NE	NE	NE	NE	NE	NE	NE	NE
	Ninth	1 (3.8)																
	Tenth	0 (0)	ES	NE	NE	NE	NE	NE	NE	NE	NE	NE	NE	NE	NE	NE	NE	NE
	Eleventh	2 (7.7)																
	Twelfth	1 (3.8)																
Parents’ Occupation	Self-employed	2 (7.7)	Statistic value	0.653 c	4.86 d	4.03 d	1.41 c	0.275 d	0.146 c	0.147 c	7.25 c	3.14 c	1.55 c	10.74 c	1.48 d	0 d	0.377 c	1.72 c
	Employee	4 (15.4)	*p*	0.554	0.088	0.133	0.316	0.871	0.867	0.866	0.123	0.116	0.287	0.010	0.477	1	0.701	0.288
	Homemaker	3 (11.5)	ES	0.165	0.6072	0.5036	0.562	0.0344	0.165	0.165	NE	−0.833	−0.528	1.613	0.1850	0	NE	0.222

Note: AN-D = Anxious Depressed; AI-D = Isolated Depressed; QS = Somatic Complaints; PI = Internalizing Problems; RR = Rule Breaking; CA = Aggressive Behavior; PE = Externalizing Problem; ES = Effect Size. a: The comparison statistic used was Welch’s *t*-test, and the effect size measure was Cohen’s *d*. b: The comparison statistic used was Mann–Whitney U, and the effect size measure was biserial correlation by ranks. c = The comparison statistic used was ANOVA and the effect size was omega squared (ω^2^). d = The comparison statistic used was Kruskal–Wallis and the effect size was epsilon squared (ε^2^). NE = Not estimable due to insufficient observations.

**Table 2 ijerph-22-00932-t002:** Descriptive Scores for CBCL and YSR for the Total Sample and by Nationality and comparison of CBCL and YSR subscales between the Mexican and Colombian samples.

Variable		M			SD			Comparison Analysis		
		Mex	Col	N	Mex	Col	N	Statistic Value	*p*	ES
CBCL	AN-D	9.63	11	10.6	5.24	5.35	5.24	−0.608 a	0.276	0.260
	AI-D	5.13	6.65	6.16	2.95	3.77	3.54	52.5 b	0.187	0.228
	QS	3.88	7.18	6.12	2.42	4.17	3.97	37 b	0.037	0.456
	PI	18.6	24.8	22.8	7.91	11.7	10.9	−1.555 a	0.068	0.620
	RR	2	5.88	4.64	1.20	4.61	4.24	39 b	0.046	0.426
	CA	8.13	10.9	10	4.64	5.80	5.52	−1.276 a	0.110	0.525
	PE	10.1	16.8	14.6	5.38	9.90	9.16	−2.167 a	0.021	0.833
YSR	AN-D	12.4	13.2	13	2.33	5.23	4.47	−0.569 a	0.287	0.213
	AI-D	5.63	6.82	6.44	3.2	3.3	3.25	−0.864 a	0.201	0.368
	QS	6	6.76	6.52	4.6	4.05	4.15	−0.402 a	0.347	0.176
	PI	24	26.8	25.9	7.23	10.3	9.37	−0.790 a	0.220	0.317
	RR	3.63	5.88	5.16	2.67	3.74	3.54	43 b	0.075	0.368
	CA	10.5	11.8	11.4	9.47	4.60	6.38	39.5 b	0.051	0.419
	PE	14.1	17.6	16.5	10.3	7.47	8.43	−0.866 a	0.203	0.391
EAT-26		10.6	14.6	13.4	6.92	9.10	8.58	−1.173 a	0.130	0.497

Note: M = Mean; SD = Standard Deviation; AN-D = Anxious Depressed; AI-D = Isolated Depressed; QS = Somatic Complaints; PI = Internalizing Problems; RR = Rule Breaking; CA = Aggressive Behavior; PE = Externalizing Problems; ES = Effect Size. a: The comparison statistic used was Welch’s *t*-test, and the effect size measure was Cohen’s *d*. b: The comparison statistic used was Mann–Whitney U, and the effect size measure was biserial correlation by ranks.

**Table 3 ijerph-22-00932-t003:** Frequency and percentage at each risk level for internalizing and externalizing problems according to the CBCL and YSR for the total sample and by nationality.

	CBCL								YSR						
Nac	RL	AN-D	AI-D	QS	PI	RR	CA	PE	AN-D	AI-D	QS	PI	RR	CA	PE
Col	NR	5 (20)	7 (28)	6 (24)	3 (12)	9 (36)	9 (36)	7 (28)	2 (8)	7 (28)	2 (8)	1 (4)	8 (32)	5 (20)	1 (4)
	AR	4 (16)	1 (4)	2 (8)	1 (4)	5 (20)	5 (20)	1 (4)	5 (20)	6 (24)	3 (12)	0 (0)	5 (20)	10 (40)	2 (8)
	CL	8 (32)	9 (36)	9 (36)	13 (52)	3 (12)	3 (12)	9 (36)	10 (40)	4 (16)	12 (48)	16 (64)	4 (16)	2 (8)	14 (56)
Mex	NR	3 (12)	5 (20)	5 (20)	2 (8)	8 (32)	7 (28)	4 (16)	0 (0)	4 (16)	1 (4)	0 (0)	7 (28)	4 (16)	0 (0)
	AR	3 (12)	1 (4)	1 (4)	1 (4)	0 (0)	1 (4)	3 (3)	2 (8)	2 (8)	4 (16)	0 (0)	0 (0)	3 (12)	4 (16)
	CL	2 (8)	2 (8)	2 (8)	5 (20)	0 (0)	0 (0)	1 (4)	6 (24)	2 (8)	3 (12)	8 (32)	1 (4)	1 (4)	4 (16)
Total N	NR	8 (32)	12 (48)	11 (44)	5 (20)	17 (68)	16 (64)	11 (44)	2 (8)	11 (44)	3 (12)	1 (4)	15 (60)	9 (36)	1 (4)
	AR	7 (28)	2 (8)	3 (12)	2 (8)	5 (20)	6 (24)	4 (16)	7 (28)	8 (32)	7 (28)	0 (0)	5 (20)	13 (52)	6 (24)
	CL	10 (40)	11 (44)	11 (44)	18 (72)	3 (12)	3 (12)	10 (40)	16 (64)	6 (24)	15 (60)	2 (24)	5 (20)	3 (12)	18 (72)

Note: RL = Risk Level; NR = No Risk; AR = At Risk; CL = Clinical; AN-D = Anxious Depressed; AI-D = Isolated Depressed; QS = Somatic Complaints; PI = Internalizing Problems; RR = Rule Breaking; CA = Aggressive Behavior; PE = Externalizing Problems.

**Table 4 ijerph-22-00932-t004:** Interaction model between risk levels of Anxious/Depressed and nationality in the EAT score.

	Sum of Squares	df	Mean Square	*F*	*p*	ω^2^
Global Model	703	4	175.7	3.37	0.030	
Anxious/Depressed Risk Levels YSR	206	2	103.0	1.97	0.166	0.058
Nationality	124	1	124.3	2.38	0.139	0.041
Anxious/Depressed Risk Levels YSR ✻ Nationality	372	1	372.5	7.14	0.015	0.183
Residuals	991	19	52.2			

## Data Availability

The dataset collected and analyzed during the current study is not publicly available, as this could violate the privacy of participants, given that it is from a clinical sample of patients with eating disorders. The corresponding author can be contacted with any relevant questions about the dataset.

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
