# Peer review of "Internalizing and Externalizing Behaviors: A Cross-Cultural Study in Colombian and Mexican Adolescents with Eating Disorders"

_ijerph, 2025, doi:10.3390/ijerph22060932_

Round 1
Reviewer 1 Report
Comments and Suggestions for Authors
Introduction
- First paragraph: for the first sentence, you need to provide some support for this statement preferably by providing some statistics from a worldwide standpoint. There may be some systematic reviews which give estimates of the worldwide burden of eating, emotional, and behavioral disorders.
- Second paragraph: I do not like the way this discussion is organized. You need to better link ASEBA to internalizing and externalizing problems rather than simply stating that certain factors fall under internalizing and externalizing problems. Please define internalizing and externalizing problems.
- Third paragraph: the first sentence belongs in the last sentence of the previous paragraph to bridge the second and third paragraphs.
- Fourth paragraph: first two sentences need to be combined since all the studies are systematic reviews estimating the prevalence of anxiety and depression. The third sentence needs more specificity pertaining to the category of externalizing problems reflected in the 14.34% figure.
- Final paragraph: your hypotheses are confusing. You state “(H1) There are statistically significant differences in internalizing and externalizing behaviors among adolescents with EDs in Colombia and Mexico. (H2) EDs are more strongly associated with internalizing behaviors than
externalizing behaviors. (H3) There are differences in the reports of Colombian and Mexican parents and adolescents regarding internalizing, externalizing, and eating problems.” For H1, I am not sure what your hypothesis is testing. Are you comparing internalizing and externalizing behaviors of adolescents with EDs across the two countries? You need to make this more clear. Additionally, I do not understand the rationale you have for H2. There is nothing in the introduction which prepares us for your comparison of internalizing and externalizing behaviors. What is the purpose of comparing these two sets of factors? This needs to be explained. As for H3, I am not sure what you are comparing. Are you comparing parents vs adolescent reports or are you comparing Colombian and Mexican reports? There is too much ambiguity in your hypotheses.
- Overall: this introduction is dilapidated and disorganized. You need to discuss eating disorders from a global perspective and then move into Latin America. Moreover, you need to explain why you chose to focus on Colombia and Mexico specifically. What makes these countries in particular worth studying? You also need to remove the discussion on Peru since it distracts from your focus on Mexico and Colombia.
Methods
- You stated “Differences between a Colombian and a Mexican sample were examined while exploring the relationships between the various analysis variables.” Please specify the analysis variables.
- Materials and Methods: It seems like you are describing an observational, cross-sectional study. You need to incorporate this language when describing your study.
- How did you get ethical approval for this study? Was there some kind of institutional review board? Please provide this information.
- Participants: Please define incidental convenience sampling method and provide a citation. Additionally, describe the setting(s) from which you obtained your convenience sampling.
- Need information about why you chose this particular study setting and what makes it suitable for your study.
- The following information does not belong in a methods section: “A total of 25 adolescents (2 males and 23 females) aged 12 to 18 years were evaluated (Males: M=13; SD=1.41; Females: M=15.3; 124 SD=1.66).” Your sample characteristics belong in the Results section.
- Your analysis plan was well-written. You clearly have a strong quantitative background.
- Overall: This section could use better organization and inclusion of key details. It ended strong with the analysis plan.
Results
- Please organize your results section by each of the hypotheses. Discuss how each set of results answers each of your hypothesis questions. Your hypotheses were not clear which made this section difficult to follow.
- You need to include a table on sample characteristics for the Colombian and Mexican samples here.
- Overall: Hard to follow due to the opaque hypotheses in the introduction section.
Discussion and Implications
- I like how you discuss the extent to which the results aligned with your hypotheses and provide adequate supporting details from the literature to amplify the context of your study. I still do not understand the point of examining Hypothesis 2. There needs to be a stronger justification in your introduction section as to why you are exploring this hypothesis.
- You should include limitations of your study in this section. The most glaring limitation is the limited external validity due to the convenience sampling in both countries which makes it difficult to generalize the results to either country or even the region.
- Overall: Missing key details and it suffers due to the poor organization of the introduction section.
Author Response
Introduction
- First paragraph: for the first sentence, you need to provide some support for this statement preferably by providing some statistics from a worldwide standpoint. There may be some systematic reviews which give estimates of the worldwide burden of eating, emotional, and behavioral disorders.
Answer: The paragraph was reorganized and supported as follows:
Eating, emotional, and behavioral disorders are among the mental health issues that became most evident following the COVID-19 pandemic. In this sense, several studies have reported a prevalence of emotional symptoms, which range between 22% and 24.9% for anxiety and 19.7% to 29% for depression. [1,2,3,4]. In another study, behavior problems were reported in 14.34% of cases for antisocial behavior and 5% for criminal behavior [5]. A systematic review carried out on 42 studies worldwide found that in 17% of the research analyzed, fears and stress derived from the pandemic increased eating disorders.These problems stem from factors such as the loss and health impacts on family and friends, an increase in domestic violence and child abuse, financial difficulties experienced by parents, and social distancing, among others (Pan American Health Organization [6].
Second paragraph: I do not like the way this discussion is organized. You need to better link ASEBA to internalizing and externalizing problems rather than simply stating that certain factors fall under internalizing and externalizing problems. Please define internalizing and externalizing problems.
- Answer: The explanation was expanded
One of the most empirically supported perspectives on emotional and behavioral problems in adolescents is the Achenbach System of Empirically Based Assessment (ASEBA). In this system, it has been possible to determine, through multivariate analysis, syndromes that group clinical problems that co-occur in a dimensional manner. [7]. This system classifies anxious/depressive symptoms withdrawal/depressive symptoms, and somatic complaints under the broad category of broadband internalizing problems that are expressed as overcontrolled patterns of emotional and affective behavior; while aggressive and rule-breaking behaviors are grouped under externalizing broadband problems, which are underregulated disruptive behavior patterns [8].
In relation to these psychological problems in adolescents, underlying transdiagnostic processes have been found, such as negative affectivity, impulsivity, cognitive biases, and difficulties in emotional regulation, which makes them more likely to show risky behaviors for their mental health ([9].
- Third paragraph: the first sentence belongs in the last sentence of the previous paragraph to bridge the second and third paragraphs.
Answer: The paragraphs were integrated.
- Fourth paragraph: first two sentences need to be combined since all the studies are systematic reviews estimating the prevalence of anxiety and depression. The third sentence needs more specificity pertaining to the category of externalizing problems reflected in the 14.34% figure.
Response: The paragraph was adjusted and integrated with the first to give it greater order as suggested by the evaluator.
Final paragraph: your hypotheses are confusing. You state “(H1) There are statistically significant differences in internalizing and externalizing behaviors among adolescents with EDs in Colombia and Mexico. (H2) EDs are more strongly associated with internalizing behaviors than externalizing behaviors. (H3) There are differences in the reports of Colombian and Mexican parents and adolescents regarding internalizing, externalizing, and eating problems.” For H1, I am not sure what your hypothesis is testing. Are you comparing internalizing and externalizing behaviors of adolescents with EDs across the two countries? You need to make this more clear. Additionally, I do not understand the rationale you have for H2. There is nothing in the introduction which prepares us for your comparison of internalizing and externalizing behaviors. What is the purpose of comparing these two sets of factors? This needs to be explained. As for H3, I am not sure what you are comparing. Are you comparing parents vs adolescent reports or are you comparing Colombian and Mexican reports? There is too much ambiguity in your hypotheses.
Answer:
The justification for each hypothesis was expanded.
- Overall: this introduction is dilapidated and disorganized. You need to discuss eating disorders from a global perspective and then move into Latin America. Moreover, you need to explain why you chose to focus on Colombia and Mexico specifically. What makes these countries in particular worth studying? You also need to remove the discussion on Peru since it distracts from your focus on Mexico and Colombia.
Answer: Information on EDs was discussed from a global perspective and then from Latin America. Information from the Peruvian study was also removed.
Methods
- You stated “Differences between a Colombian and a Mexican sample were examined while exploring the relationships between the various analysis variables.” Please specify the analysis variables.
Answer: The variables that were compared were specified (internalizing behaviors, externalizing behaviors, and eating problems)
- Materials and Methods: It seems like you are describing an observational, cross-sectional study. You need to incorporate this language when describing your study.
Answer: The information suggested by the evaluator was included: This study is a cross-sectional observational study with an associative strategy to conduct a comparative and predictive study.
- How did you get ethical approval for this study? Was there some kind of institutional review board? Please provide this information.
Answer: The procedure specified the following information: In the first phase, the study was approved by the University Ethics Committee that led the study according to Minutes No. 33 of October 17, 2023.
- Participants: Please define incidental convenience sampling method and provide a citation. Additionally, describe the setting(s) from which you obtained your convenience sampling. Need information about why you chose this particular study setting and what makes it suitable for your study.
Answer: The information was completed as follows: An incidental convenience sampling method was used, which refers to the process of selecting a representative sample that meets the necessary characteristics for the research [32].
Participants were also to receive treatment for eating disorders in private clinics specializing in the clinical care of these problems, one in Bogotá and another in Mexico City.
- The following information does not belong in a methods section: “A total of 25 adolescents (2 males and 23 females) aged 12 to 18 years were evaluated (Males: M=13; SD=1.41; Females: M=15.3; 124 SD=1.66).” Your sample characteristics belong in the Results section.
Answer: This information has been relocated to the results section.
- Your analysis plan was well-written. You clearly have a strong quantitative background.
- Overall: This section could use better organization and inclusion of key details. It ended strong with the analysis plan.
Results
- Please organize your results section by each of the hypotheses. Discuss how each set of results answers each of your hypothesis questions. Your hypotheses were not clear which made this section difficult to follow.
Answer: The analysis of results was detailed in response to each of the hypotheses.
- You need to include a table on sample characteristics for the Colombian and Mexican samples here.
Answer: We have now included a table presenting the sociodemographic characteristics of the participants, separated by country (Colombia and Mexico), as recommended.
Table 1.
Description of the sociodemographic characteristics of the samples for each country.
|
Sociodemographic characteristics |
Colombian |
Mexican |
|
|
Age |
Mean |
15.4 |
14.1 |
|
Standard Deviation |
1.80 |
2.15 |
|
|
Gender |
Male |
2 (7.7) |
0 (0) |
|
Female |
15 (57.7) |
9 (34.6) |
|
|
Parents' Gender |
Male |
5 (19.2) |
2 (7.7) |
|
Female |
12 (46.2) |
7 (26.9) |
|
|
Grade |
Sixth |
1 (3.8) |
1 (3.8) |
|
Seventh |
2 (7.7) |
1 (3.8) |
|
|
Eighth |
0 (0) |
3 (11.5) |
|
|
Ninth |
5 (19.2) |
1 (3.8) |
|
|
Tenth |
3 (11.5) |
0 (0) |
|
|
Eleventh |
6 (23.1) |
2 (7.7) |
|
|
Twelfth |
0 (0) |
1 (3.8) |
|
|
Parents' Occupation |
Self-employed |
4 (15.4) |
2 (7.7) |
|
Employee |
12 (46.2) |
4 (15.4) |
|
|
Homemaker |
0 (0) |
3 (11.5) |
|
|
Other |
1 (3.8) |
0 (0) |
|
- Overall: Hard to follow due to the opaque hypotheses in the introduction section.
Discussion and Implications
- I like how you discuss the extent to which the results aligned with your hypotheses and provide adequate supporting details from the literature to amplify the context of your study. I still do not understand the point of examining Hypothesis 2. There needs to be a stronger justification in your introduction section as to why you are exploring this hypothesis.
Answer: The justification for hypothesis 2 was expanded
- You should include limitations of your study in this section. The most glaring limitation is the limited external validity due to the convenience sampling in both countries which makes it difficult to generalize the results to either country or even the region. \
Answer: The explanation of the limitations was expanded as suggested by the evaluator.
The limitations of this study lie in the small and unequal sample sizes between Mexico and Colombia and the convenience sampling in both countries, making it difficult to generalize the results to either country or even the region.
- Overall: Missing key details and it suffers due to the poor organization of the introduction section.
Answer: The introduction section was reorganized

Reviewer 2 Report
Comments and Suggestions for Authors
The article concerns very important issues that should be researched and described in scientific literature. Additionally, the cross-cultural analyses on functioning of adolescents are very desirable, so those are the merits of the presented articles.
At the same time in the text certain problems can be observed. First of all, the theoretical introduction of the article has to be re-written. The Authors have to be specific about introducing the topic of internalizing and externalizing behaviors in persons suffering eating disorders.
Since the cross-cultural comparisons are planned it is needed to give additional description of cross-cultural research on that matter, and to explain the cultural differences between Columbia and Mexico.
The text concerns adolescents therefore special attention should be given to the functioning (in the light of externalizing and internalizing behaviors) of Colombian and Mexican teenagers.
Also, since Authors compare self-reports and other reports it is needed to give a thorough description of the differences between self-reports and parents' reports for psychological properties observed in adolescents.
Second of all, the procedure of the study has to be described in more detail. How and based on which criteria eating disorders were diagnosed? What type of eating disorders were identified in respondents? Where the respondents were recruited?
Additionally, Authors have interpreted some descriptive statistics without any additional tests. You shouldn't do that. If you are interested in comparison of the mean levels of specific variables you have to conduct specific statistical tests instead of only looking at the level of mean for this variable.
The data presented in the article is not sufficiently analyzed. It needs additional analyses and more thorough description.
Thirdly, the discussion has to be re-written. Currently it includes misinterpretations of the data given in the article (e.g. cause and effect argumentation for correlational data).
Additionally, the discussion lacks thorough interpretation of the results in the light of previous studies and is mostly 'technical'.
Also the conclusions presented in the text are not in correspondence to the data presented in the article.
To summarize, the text concerns a very interesting topic and after major changes should be published in the journal.
Among the advantages of the proposed articles are: respondents groups, cross-cultural data, and results gathered from adolescents and their parents. In order for the readers to fully appreciate abovementioned advantages Authors have to modify specific sections of the article so the data will be clearly described, analyzed and interpreted.
Author Response
The article concerns very important issues that should be researched and described in scientific literature. Additionally, the cross-cultural analyses on functioning of adolescents are very desirable, so those are the merits of the presented articles.
At the same time in the text certain problems can be observed. First of all, the theoretical introduction of the article has to be re-written. The Authors have to be specific about introducing the topic of internalizing and externalizing behaviors in persons suffering eating disorders.
Answer: In the introduction, the theoretical basis on internalizing and externalizing in adolescents with eating disorders problems was expanded and adjusted.
Since the cross-cultural comparisons are planned it is needed to give additional description of cross-cultural research on that matter, and to explain the cultural differences between Columbia and Mexico.
Answer: In the introduction, more information was provided:
The tripartite model of EDs posits that family, friends, and the media exert a dysfunctional influence on body image and internalizing symptoms in adolescents with EDs [29]. However, a study conducted in Mexico found that family pressure had a more significant impact than peer and media pressure on the body image of adolescents with EDs through negative messages, eating behaviors, and home-based diets. These factors may be associated with the fact that in Latin American contexts, families, especially mothers, tend to have a significant influence on their children's eating behaviors [30].
In the Colombian cultural context, a relationship has been documented between a history of eating disorders in parents, criticism and jokes about their children's body shape and weight, and eating problems in adolescents, which ends up leading them to associate success and social approval with thinness, especially in women [18]. Adolescents with EDs are among the most vulnerable groups for mental health problems, given their heightened risks of emotional and behavioral issues. Paradoxically, they are one of the least studied groups, particularly in the Latin American context.
Furthermore, there has been a lack of more studies with clinical samples, both in Colombia and Mexico, that account for the relationships between internalizing and externalizing problems and EDs from the perspective of adolescents and their parents, especially if one takes into account that some of the findings presented have reported variability on the association between these variables in samples of school-aged adolescents [18,19,20,24].
The text concerns adolescents therefore special attention should be given to the functioning (in the light of externalizing and internalizing behaviors) of Colombian and Mexican teenagers.
Answer: In the introduction, more information was provided:
In relation to these psychological problems in adolescents, underlying transdiagnostic processes have been found, such as negative affectivity, impulsivity, cognitive biases, and difficulties in emotional regulation, which makes them more likely to show risky behaviors for their mental health ([9].
Also, since Authors compare self-reports and other reports it is needed to give a thorough description of the differences between self-reports and parents' reports for psychological properties observed in adolescents.
Answer: This information can be found in the introduction:
limited parental and peer connections are associated with externalizing and internalizing problems and EDs in adolescents [26,27]. Parents' difficulties in identifying their children's emotional and eating problems have also been documented, which is reflected in a discrepancy in the reporting of these problems between adolescents and their parents limiting the seeking of timely treatment for their children [28].
Second of all, the procedure of the study has to be described in more detail. How and based on which criteria eating disorders were diagnosed? What type of eating disorders were identified in respondents? Where the respondents were recruited?
Answer: Participants were also required to be receiving treatment for an eating disorder such as anorexia, bulimia, or binge eating at private clinics specializing in these conditions, one in Bogotá and one in Mexico City.
Additionally, Authors have interpreted some descriptive statistics without any additional tests. You shouldn't do that. If you are interested in comparison of the mean levels of specific variables you have to conduct specific statistical tests instead of only looking at the level of mean for this variable.
Answer: The data were analyzed both descriptively and inferentially, based on the distribution of the variables and the sample sizes.
The data presented in the article is not sufficiently analyzed. It needs additional analyses and more thorough description.
Answer: The data were analyzed both descriptively and inferentially, based on the distribution of the variables and the sample sizes.
Thirdly, the discussion has to be re-written. Currently it includes misinterpretations of the data given in the article (e.g. cause and effect argumentation for correlational data).
Answer: It was verified that no interpretations of causality were made in the discussion.
Additionally, the discussion lacks thorough interpretation of the results in the light of previous studies and is mostly 'technical'.
Answer: The discussion was extended
Also the conclusions presented in the text are not in correspondence to the data presented in the article.
Answer: It was verified that the conclusions presented in the text corresponded to the data presented in the article.
These results emphasize the need for early detection through a multicomponent assessment that includes the participation of parents and peers. However, special attention should be paid to adolescents' self-reports, as they reported a higher percentage of internalizing problems than their parents in this study. This approach may help prevent psychological problems, particularly internalizing disorders, and address the behavioral consequences of weight-related stigma and bias.
It is important to note that eating disorders in the assessed samples from Colombia and Mexico were more associated with internalizing anxiety-depressive problems than with externalizing behaviors, which would indicate a tendency for these adolescents to exhibit emotional overregulation, cognitive biases, impulsivity, and greater negative affectivity. However, the small and unequal sample sizes between Mexico and Colombia limit the generalizability of the findings.
To summarize, the text concerns a very interesting topic and after major changes should be published in the journal.
Among the advantages of the proposed articles are: respondents groups, cross-cultural data, and results gathered from adolescents and their parents. In order for the readers to fully appreciate abovementioned advantages Authors have to modify specific sections of the article so the data will be clearly described, analyzed and interpreted.

Reviewer 3 Report
Comments and Suggestions for Authors
Introduction:
- Authors need to explore further internalising and externalising in EDs: the available studies on pathomechanism, correlation with the cultures studied (Mexico and Colombia) with body weight and food anxiety, the impact of globalisation (social media, urbanisation) on internalising and externalising, and individual responses to these behaviours across different age categories (e.g., children, adolescents).
- Information on the epidemiological studies can be reduced.
Methods:
- Please include the information on which forms were filled out by parents and which forms were filled out by their parents.
- Please provide the ethical approval details and study registration.
Results:
- Tables 1&2 are a little hard to read. I argue skewness and kurtosis need to be reported in this paper. Please transpose the columns and rows and merge them with Table 3 for the effect sizes. I would suggest looking at Table 2 of this example: https://doi.org/10.1186/s40337-020-00337-w
- Where is the sociodemographic data?
Discussion:
- I would suggest including a figure or conceptual model on how these findings align with the existing theories and can improve the current practice. Example of figure: (figure 1) DOI:1007/s00127-014-1004-z
- More exploration of cultural differences and the results are needed to align the discussion with the title, e.g. body shape and body type point of view, social norms,
- Please provide the strengths and limitations of the study
- Further suggestions in the results and discussions are pending, as the sociodemographic data was not found in the article.
Conclusion:
- Please rewrite the conclusion. I noticed it is a copy of the discussion from lines 264-269 and lines 276-281. A conclusion should provide a summary of the study and future recommendations.
Minor comment:
- There is a typing mistake in line 39.
Author Response
Introduction:
- Authors need to explore further internalising and externalising in EDs: the available studies on pathomechanism, correlation with the cultures studied (Mexico and Colombia) with body weight and food anxiety, the impact of globalisation (social media, urbanisation) on internalising and externalising, and individual responses to these behaviours across different age categories (e.g., children, adolescents).
- Answer: The explanation was expanded
One of the most empirically supported perspectives on emotional and behavioral problems in adolescents is the Achenbach System of Empirically Based Assessment (ASEBA). In this system, it has been possible to determine, through multivariate analysis, syndromes that group clinical problems that co-occur in a dimensional manner. [7]. This system classifies anxious/depressive symptoms withdrawal/depressive symptoms, and somatic complaints under the broad category of broadband internalizing problems that are expressed as overcontrolled patterns of emotional and affective behavior; while aggressive and rule-breaking behaviors are grouped under externalizing broadband problems, which are underregulated disruptive behavior patterns [8].
In relation to these psychological problems in adolescents, underlying transdiagnostic processes have been found, such as negative affectivity, impulsivity, cognitive biases, and difficulties in emotional regulation, which makes them more likely to show risky behaviors for their mental health ([9].
- Information on the epidemiological studies can be reduced.
Answer: The paragraph was reorganized and supported as follows:
Eating, emotional, and behavioral disorders are among the mental health issues that became most evident following the COVID-19 pandemic. In this sense, several studies have reported a prevalence of emotional symptoms, which range between 22% and 24.9% for anxiety and 19.7% to 29% for depression. [1,2,3,4]. In another study, behavior problems were reported in 14.34% of cases for antisocial behavior and 5% for criminal behavior [5]. A systematic review carried out on 42 studies worldwide found that in 17% of the research analyzed, fears and stress derived from the pandemic increased eating disorders.These problems stem from factors such as the loss and health impacts on family and friends, an increase in domestic violence and child abuse, financial difficulties experienced by parents, and social distancing, among others (Pan American Health Organization [6].
Methods:
- Please include the information on which forms were filled out by parents and which forms were filled out by their parents.
Answer: This information was specified in the procedure section.
The researchers created an online version of the EAT-26 questionnaire and the YSR to adolescents and administered the CBCL to parents. These instruments were administered in person at the clinics where adolescents received treatment.
- Please provide the ethical approval details and study registration.
Answer: The procedure specified the following information: In the first phase, the study was approved by the University Ethics Committee that led the study according to Minutes No. 33 of October 17, 2023.
Results:
- Tables 1&2 are a little hard to read. I argue skewness and kurtosis need to be reported in this paper. Please transpose the columns and rows and merge them with Table 3 for the effect sizes. I would suggest looking at Table 2 of this example: https://doi.org/10.1186/s40337-020-00337-w
- Where is the sociodemographic data?
Answer: A new Table 1 has been added, presenting the sociodemographic description of the samples. As for the skewness and kurtosis data, these were already reported in the table titled Descriptive Scores for CBCL and YSR for the Total Sample and by Nationality. However, what are now presented in Table 2 are the merged versions of the former Tables 1 and 3, including descriptive statistics for each sample and the total sample, as well as the comparison statistics, as suggested.
Table 1.
Description of the sociodemographic characteristics of the samples for each country.
|
Sociodemographic characteristics |
Colombian |
Mexican |
|
|
Age |
Mean |
15.4 |
14.1 |
|
Standard Deviation |
1.80 |
2.15 |
|
|
Gender |
Male |
2 (7.7) |
0 (0) |
|
Female |
15 (57.7) |
9 (34.6) |
|
|
Parents' Gender |
Male |
5 (19.2) |
2 (7.7) |
|
Female |
12 (46.2) |
7 (26.9) |
|
|
Grade |
Sixth |
1 (3.8) |
1 (3.8) |
|
Seventh |
2 (7.7) |
1 (3.8) |
|
|
Eighth |
0 (0) |
3 (11.5) |
|
|
Ninth |
5 (19.2) |
1 (3.8) |
|
|
Tenth |
3 (11.5) |
0 (0) |
|
|
Eleventh |
6 (23.1) |
2 (7.7) |
|
|
Twelfth |
0 (0) |
1 (3.8) |
|
|
Parents' Occupation |
Self-employed |
4 (15.4) |
2 (7.7) |
|
Employee |
12 (46.2) |
4 (15.4) |
|
|
Homemaker |
0 (0) |
3 (11.5) |
|
|
Other |
1 (3.8) |
0 (0) |
|
Table 2. Descriptive Scores for CBCL and YSR for the Total Sample and by Nationality and comparison of CBCL and YSR subscales between the Mexican and Colombian samples.
|
Variable |
M |
DE |
S |
K |
p - SW |
Comparison analysis |
|||||||||||||
|
Mex |
Col |
N |
Mex |
Col |
N |
Mex |
Col |
N |
Mex |
Col |
N |
Mex |
Col |
N |
Statistic value |
p |
ES |
||
|
CBCL |
AN-D |
9.63 |
11 |
10.6 |
5.24 |
5.35 |
5.24 |
1.04 |
.30 |
.47 |
.069 |
-.91 |
-.91 |
.17 |
.63 |
.13 |
-.608a |
.276 |
.260 |
|
AI-D |
5.13 |
6.65 |
6.16 |
2.95 |
3.77 |
3.54 |
.24 |
-.092 |
.101 |
-1.29 |
-1.72 |
-1.52 |
.45 |
.011 |
.01 |
52.5b |
.187 |
.228 |
|
|
QS |
3.88 |
7.18 |
6.12 |
2.42 |
4.17 |
3.97 |
.11 |
.099 |
.47 |
-1.47 |
-1.43 |
-.96 |
.27 |
.12 |
.03 |
37 b |
.037 |
.456 |
|
|
PI |
18.6 |
24.8 |
22.8 |
7.91 |
11.7 |
10.9 |
.11 |
.045 |
.29 |
-1.29 |
-1.42 |
-1.09 |
.75 |
.31 |
.21 |
-1.555 a |
.068 |
.620 |
|
|
RR |
2 |
5.88 |
4.64 |
1.20 |
4.61 |
4.24 |
0 |
.433 |
.99 |
.81 |
-.84 |
.051 |
.53 |
.15 |
.004 |
39 b |
.046 |
.426 |
|
|
CA |
8.13 |
10.9 |
10 |
4.64 |
5.80 |
5.52 |
.16 |
.11 |
.25 |
.14 |
-.76 |
-.58 |
.96 |
.25 |
.35 |
-1.276 a |
.110 |
.525 |
|
|
PE |
10.1 |
16.8 |
14.6 |
5.38 |
9.90 |
9.16 |
.22 |
.214 |
.58 |
-.60 |
-1.11 |
-.58 |
.79 |
.22 |
.12 |
-2.167 a |
.021 |
.833 |
|
|
YSR |
AN-D |
12.4 |
13.2 |
13 |
2.33 |
5.23 |
4.47 |
-1.06 |
-.154 |
-.045 |
.49 |
-.94 |
-.35 |
.31 |
.43 |
.85 |
-.569 a |
.287 |
.213 |
|
AI-D |
5.63 |
6.82 |
6.44 |
3.2 |
3.3 |
3.25 |
.63 |
-.284 |
-.019 |
-.86 |
-.24 |
-.75 |
.48 |
.61 |
.48 |
-.864 a |
.201 |
.368 |
|
|
QS |
6 |
6.76 |
6.52 |
4.6 |
4.05 |
4.15 |
1.36 |
.13 |
.492 |
1.14 |
-.86 |
0.69 |
.027 |
.79 |
.14 |
-.402 a |
.347 |
.176 |
|
|
PI |
24 |
26.8 |
25.9 |
7.23 |
10.3 |
9.37 |
.44 |
-.691 |
-.41 |
-1.03 |
.12 |
-.16 |
.62 |
.48 |
.57 |
-.790 a |
.220 |
.317 |
|
|
RR |
3.63 |
5.88 |
5.16 |
2.67 |
3.74 |
3.54 |
.90 |
1.11 |
1.16 |
2.23 |
2.22 |
2.34 |
.14 |
.19 |
.04 |
43 b |
.075 |
.368 |
|
|
CA |
10.5 |
11.8 |
11.4 |
9.47 |
4.60 |
6.38 |
2.39 |
.202 |
1.70 |
6.22 |
.44 |
4.62 |
.002 |
.98 |
.004 |
39.5 b |
.051 |
.419 |
|
|
PE |
14.1 |
17.6 |
16.5 |
10.3 |
7.47 |
8.43 |
1.89 |
.19 |
.72 |
3.77 |
-.51 |
.046 |
.019 |
.92 |
.15 |
-.866 a |
.203 |
.391 |
|
|
EAT-26 |
10.6 |
14.6 |
13.4 |
6.92 |
9.10 |
8.58 |
1.18 |
.60 |
.76 |
.40 |
.30 |
.35 |
.16 |
.44 |
.13 |
-1.173 a |
.130 |
.497 |
|
- Note: M = Mean; DE = Standard Deviation; S = Skewness; K = Kurtosis; p-SW = p-value of Shapiro-Wilk test; AN-D = Anxious Depressed; AI-D = Isolated Depressed; QS = Somatic Complaints; PI = Internalizing Problems; RR = Rule Breaking; CA = Aggressive Behavior; PE = Externalizing Problems."; ES= Effect Size. a: The comparison statistic used was Welch's t-test, and the effect size measure was Cohen's d. b: The comparison statistic used was Mann-Whitney U, and the effect size measure was Biserial correlation by ranks.
Discussion:
- I would suggest including a figure or conceptual model on how these findings align with the existing theories and can improve the current practice. Example of figure: (figure 1) DOI:1007/s00127-014-1004-z
Answer: The figure 1 with the model was included at the end of the introduction.
- More exploration of cultural differences and the results are needed to align the discussion with the title, e.g. body shape and body type point of view, social norms,
Answer: The explanation on these aspects was expanded.
These findings show partial support for hypothesis 1 and show the recognition of these problems by adolescents as has been shown in other studies with Latin American adolescents [11,20,21]. These results are possibly linked to a greater emotional sensitivity of these adolescents to criticism about their body image in the family environment and to a need for social approval [18].
These findings are consistent with those of Moreno Encinas et al. [23] and suggest that internalizing problems in adolescence may be a factor associated with eating disorders. Likewise, it could be explained because adolescents with EDs the familiar and social media exerted a dysfunctional influence on body image and internalizing symptoms in adolescents with EDs [18,29].
- Please provide the strengths and limitations of the study
Answer: The following strengths and limitations were included in the conclusions:
These results emphasize the need for early detection using a multicomponent evaluation that includes input from parents, educators, and peers. However, particular attention should be given to adolescents' self-assessments, as they reported a higher percentage of internalizing problems than identified by their parents in this study. This approach can help prevent psychological problems, particularly internalizing disorders, and address behavioral consequences of weight stigma and bias [25].
The findings highlight the need to improve mental health among adolescents in both countries, consistent with Chavira et al. [44], who emphasize the importance of developing evidence-based studies to address these issues. Such research can inform funding for prevention and intervention programs and propose changes to public policies to more effectively meet the mental health needs of adolescents in the post-pandemic period, especially those at higher risk, such as those with eating disorders.
It is important to note that eating disorders are associated by other psychopathological variables, and the results related to internalization may be linked to comorbidities with anxiety and depression disorders. Future research should include a review of participants' medical histories to identify comorbidities and incorporate other diagnoses.
The limitations of this study lie in the small and unequal sample sizes between Mexico and Colombia and the convenience sampling in both countries, making it difficult to generalize the results to either country or even the region. It is important to acknowledge that the small and unequal sample sizes by country (n = 17 for Colombia, n = 8 for Mexico) represent a limitation that may have influenced the statistical power of the study. Small sample sizes make it hard to find significant effects, especially if they are small or moderate. Additionally, group size imbalance can affect the stability of estimates and limit the generalizability of the findings. In this regard, although effect sizes were reported alongside p-values, they should be interpreted with caution, as effect sizes in small samples can be unstable or overestimated [45].
Qualitative data were not obtained that would allow contextualizing the discrepancies between the reports of adolescents and parents since the clinics only allowed the application of the instruments due to confidentiality issues regarding patient information.
Future studies are needed with larger and more balanced samples across countries, complementing the collection of information from adolescents with qualitative assessment strategies that contribute to greater consistency and robustness of the findings.
- Further suggestions in the results and discussions are pending, as the sociodemographic data was not found in the article.
Answer: Information on sociodemographic variables is expanded, in addition to suggestions and conclusions.
Conclusion:
- Please rewrite the conclusion. I noticed it is a copy of the discussion from lines 264-269 and lines 276-281. A conclusion should provide a summary of the study and future recommendations.
Answer: The conclusions were adjusted
Minor comment:
- There is a typing mistake in line 39.
Answer: The error was corrected

Reviewer 4 Report
Comments and Suggestions for Authors
This study investigates internalizing and externalizing behaviors among adolescents with eating disorders (EDs) in Colombia and Mexico, using both self-reports and parental reports. The authors analyze the differences in emotional and behavioral symptoms between the two countries and between informants (parents vs. adolescents), using validated instruments (CBCL, YSR, EAT-26). Results show higher rates of internalizing symptoms and ED-related behaviors in Colombian adolescents and highlight discrepancies between parent and adolescent reports.
Some aspects are innovative and in my opinion make this study a valuable and original contribution to the scientific literature.
This study compares data between two countries and in the literature there is little scientific evidence of intercultural comparisons; also the comparison between the self-assessments of adolescents and parents to assess emotional and behavioral symptoms makes this study original. The in-depth analysis of the discrepancies between young people and parents offers interesting food for thought for early diagnosis and interventions as it suggests that young people could recognize symptoms more easily than their parents
The inclusion of externalizing behaviors is also very interesting, which together with the focus on internalizing behaviors contributes to providing a more complete vision of psychopathological aspects.
Finally, the use of internationally validated tools adapted to the study population contributes to the cultural validation of the study.
This study makes a meaningful contribution to the understanding of internalizing and externalizing symptoms in adolescents with EDs in Latin America. Its strengths lie in the use of validated instruments, multiple informants, and a cross-national design. However, small sample size and modest effect sizes limit the strength of some conclusions.
Some improvements could be made, especially in: Sample Size and Generalizability and Reporting of Statistical Analysis. Below, some questions for the Authors:
- In Introduction paragraph, the rationale for choosing Colombia and Mexico as comparison cases could be more explicitly theorized. What socio-cultural factors might explain behavioral or reporting differences? Could the Authors discuss more fully how national context, stigma, access to care, or cultural perceptions of EDs may influence self-reporting vs. parental perception?
- Could the Authors elaborate on the theoretical or socio-cultural differences between Colombia and Mexico that might explain the higher rates of internalizing and externalizing symptoms in the Colombian group?
- In Participants paragraph, the Authors used a very small sample for country-level comparisons. Could the Authors explain in the discussion how the small and unequal sample size (n=17 for Colombia, n=8 for Mexico) may have influenced the statistical power and interpretation of the effect sizes?
- In Results paragraph the Authors described briefly some results (e.g. Table 4; line 227) without detailed interpretation. Could the authors broaden the discussion on the interaction effects between the YSR Anxious/Depressed subscale x nationality, to improve understanding of the clinical implications?
- Did the Authors consider any qualitative data (e.g., interviews or clinical notes) to contextualize discrepancies between adolescent and parent reports? If they didn't consider them, can they explain the reasons?
- Were there any observed gender differences in the sample, and if so, how were they addressed or controlled for?
- What are the recommendations for clinicians or educators working with adolescents and families where these discrepancies in perceived mental health occur?
I suggest reading again the manuscript because, im my opinion, it is necessary revise phrasing for fluency and grammar, particularly in the discussion and abstract.
Author Response
Comments and Suggestions for Authors
This study investigates internalizing and externalizing behaviors among adolescents with eating disorders (EDs) in Colombia and Mexico, using both self-reports and parental reports. The authors analyze the differences in emotional and behavioral symptoms between the two countries and between informants (parents vs. adolescents), using validated instruments (CBCL, YSR, EAT-26). Results show higher rates of internalizing symptoms and ED-related behaviors in Colombian adolescents and highlight discrepancies between parent and adolescent reports.
Some aspects are innovative and in my opinion make this study a valuable and original contribution to the scientific literature.
This study compares data between two countries and in the literature there is little scientific evidence of intercultural comparisons; also the comparison between the self-assessments of adolescents and parents to assess emotional and behavioral symptoms makes this study original. The in-depth analysis of the discrepancies between young people and parents offers interesting food for thought for early diagnosis and interventions as it suggests that young people could recognize symptoms more easily than their parents
The inclusion of externalizing behaviors is also very interesting, which together with the focus on internalizing behaviors contributes to providing a more complete vision of psychopathological aspects.
Finally, the use of internationally validated tools adapted to the study population contributes to the cultural validation of the study.
This study makes a meaningful contribution to the understanding of internalizing and externalizing symptoms in adolescents with EDs in Latin America. Its strengths lie in the use of validated instruments, multiple informants, and a cross-national design. However, small sample size and modest effect sizes limit the strength of some conclusions.
Some improvements could be made, especially in: Sample Size and Generalizability and Reporting of Statistical Analysis. Below, some questions for the Authors:
- In Introduction paragraph, the rationale for choosing Colombia and Mexico as comparison cases could be more explicitly theorized. What socio-cultural factors might explain behavioral or reporting differences? Could the Authors discuss more fully how national context, stigma, access to care, or cultural perceptions of EDs may influence self-reporting vs. parental perception?
Answer: This information was expanded throughout the introduction.
- Could the Authors elaborate on the theoretical or socio-cultural differences between Colombia and Mexico that might explain the higher rates of internalizing and externalizing symptoms in the Colombian group?
Answer:
It could also be hypothesized that there was greater recognition of problems in Colombian adolescents compared to Mexican adolescents; although discrepancies in sample size could also have influenced these results, given that the number of participants evaluated in Colombia was more than double that of Mexico. These findings show partial support for hypothesis 1 and show the recognition of these problems by adolescents as has been shown in other studies with Latin American adolescents [11,20,21]. These results are possibly linked to a greater emotional sensitivity of these adolescents to criticism about their body image in the family environment and to a need for social approval [18].
In Participants paragraph, the Authors used a very small sample for country-level comparisons. Could the Authors explain in the discussion how the small and unequal sample size (n=17 for Colombia, n=8 for Mexico) may have influenced the statistical power and interpretation of the effect sizes?
Answer: The following paragraph has been included in the discussion:
The limitations of this study lie in the small and unequal sample sizes between Mexico and Colombia and the convenience sampling in both countries, making it difficult to generalize the results to either country or even the region. It is important to acknowledge that the small and unequal sample sizes by country (n = 17 for Colombia, n = 8 for Mexico) represent a limitation that may have influenced the statistical power of the study. Small sample sizes make it hard to find significant effects, especially if they are small or moderate. Additionally, group size imbalance can affect the stability of estimates and limit the generalizability of the findings. In this regard, although effect sizes were reported alongside p-values, they should be interpreted with caution, as effect sizes in small samples can be unstable or overestimated [45].
Qualitative data were not obtained that would allow contextualizing the discrepancies between the reports of adolescents and parents since the clinics only allowed the application of the instruments due to confidentiality issues regarding patient information.
Future studies are needed with larger and more balanced samples across countries, complementing the collection of information from adolescents with qualitative assessment strategies that contribute to greater consistency and robustness of the findings.
- In Results paragraph the Authors described briefly some results (e.g. Table 4; line 227) without detailed interpretation. Could the authors broaden the discussion on the interaction effects between the YSR Anxious/Depressed subscale x nationality, to improve understanding of the clinical implications?
Answer: The following paragraph has been included in the results:
This model suggests that risk assessment scores do not follow a uniform trend that can be explained solely by risk level or nationality. In other words, the relationship between risk level (no risk, at risk, clinical) and scale scores varies depending on nationality (Colombian vs. Mexican) (Table 4). These findings could indicate that Colombian adolescents with eating disorders tend to report more anxious-depressive symptoms compared to Mexican adolescents.
- Did the Authors consider any qualitative data (e.g., interviews or clinical notes) to contextualize discrepancies between adolescent and parent reports? If they didn't consider them, can they explain the reasons?
Answer: The following paragraph has been included in the limitations
Future studies are needed with larger and more balanced samples across countries, complementing the collection of information from adolescents with qualitative assessment strategies that contribute to greater consistency and robustness of the findings.
- Were there any observed gender differences in the sample, and if so, how were they addressed or controlled for?
Answer: No significant differences were established based on gender given the small number of men who participated in the study, which was suggested to be increased for future studies.
- What are the recommendations for clinicians or educators working with adolescents and families where these discrepancies in perceived mental health occur?
Answer: The following paragraph has been included in the conclusions:
These results emphasize the need for early detection through a multicomponent assessment that includes the participation of parents and peers. However, special attention should be paid to adolescents' self-reports, as they reported a higher percentage of internalizing problems than their parents in this study. This approach may help prevent psychological problems, particularly internalizing disorders, and address the behavioral consequences of weight-related stigma and bias.
Comments on the Quality of English Language
I suggest reading again the manuscript because, im my opinion, it is necessary revise phrasing for fluency and grammar, particularly in the discussion and abstract.
Response: Some adjustments were made to the translation of the entire text

Round 2
Reviewer 2 Report
Comments and Suggestions for Authors
Thank you for the changes made in the text. Unfortunately I still see the sections that need major revision. After receiving a thorough description of the research procedure I do not see any connection between hypotheses and theoretical introduction of the article. Additionally, I still do not understand why you use self-reports and parental reports. From a scientific point of view the analyses presented in the text are not reliable, since the research group was so small and consisted of persons with many different eating disorders. As I was pointing out in the previous revision each eating disorder will be different and differently corresponding to externalizing or internalizing behaviors. The study you've conducted serves as a good start for very important research and scientific paper. Please expand your research, collect more data from persons with different eating disorders diagnoses, equalize the number of participants from both countries (also taking into consideration approximately equal number of respondents with each ED diagnosis) and then analyze your data again.
Author Response
Dear reviewer
We thank you in advance for your time and valuable suggestions for improving our manuscript.
Adjustments were made in response to your recommendations.
Comments and Suggestions for Authors
Thank you for the changes made in the text. Unfortunately I still see the sections that need major revision. After receiving a thorough description of the research procedure I do not see any connection between hypotheses and theoretical introduction of the article.
Response
Regarding hypothesis 1, the following argument was included:
In Latin America, some studies have explored the relationship between internalizing and externalizing problems and EDs. However, there is no empirical evidence that explores the differences between clinical samples from these Latin American countries
Regarding hypothesis 2
Response
The following arguments were expanded in the document:
In relation to these psychological problems in adolescents, underlying transdiagnostic processes have been found, such as negative affectivity, impulsivity, cognitive biases, and difficulties in emotional regulation, which makes them more likely to show risky behaviors for their mental health ([9].
Empirical evidence indicates that adolescents with internalizing and externalizing problems frequently exhibit Eating Disorders (EDs) [10,11]. EDs are characterized by concerns about weight, height, body shape, and image, as well as disrupted eating habits [12].
In a scoping review conducted in Colombia with adolescents, a correlation was found between eating disorders of anorexia and bulimia with internalizing symptoms of an anxious-depressive type [13]. In a comparative study conducted in Mexico, it was found that the risk of developing bulimia and anorexia is higher when anxiety is high [14. These studies have suggested further research in this field, given that no clear relationship between eating disorders and externalizing problems has been demonstrated.
Response
Regarding hypothesis 3, the following arguments are found:
Furthermore, limited parental and peer connections are associated with externalizing and internalizing problems and EDs in adolescents [26,27]. Parents' difficulties in identifying their children's emotional and eating problems have also been documented, which is reflected in a discrepancy in the reporting of these problems between adolescents and their parents limiting the seeking of timely treatment for their children [28].
The tripartite model of EDs posits that family, friends, and the media exert a dysfunctional influence on body image and internalizing symptoms in adolescents with EDs [29]. However, a study conducted in Mexico found that family pressure had a more significant impact than peer and media pressure on the body image of adolescents with EDs through negative messages, eating behaviors, and home-based diets. These factors may be associated with the fact that in Latin American contexts, families, especially mothers, tend to have a significant influence on their children's eating behaviors [30].
In the Colombian cultural context, a relationship has been documented between a history of eating disorders in parents, criticism and jokes about their children's body shape and weight, and eating problems in adolescents, which ends up leading them to associate success and social approval with thinness, especially in women [18]. Adolescents with EDs are among the most vulnerable groups for mental health problems, given their heightened risks of emotional and behavioral issues. Paradoxically, they are one of the least studied groups, particularly in the Latin American context.
Furthermore, there has been a lack of more studies with clinical samples, both in Colombia and Mexico, that account for the relationships between internalizing and externalizing problems and EDs from the perspective of adolescents and their parents, especially if one takes into account that some of the findings presented have reported variability on the association between these variables in samples of school-aged adolescents [18,19,20,24].
-Additionally, I still do not understand why you use self-reports and parental reports.
Response:
Some paragraphs were included in response to reviewer 2's suggestion.
It has also been documented those adolescents with EDs report fewer externalizing behaviors, such as rule-breaking, compared to what their parents report. This discrepancy highlights the need for a more comprehensive assessment based on reports from both parents and adolescents [10].
limited parental and peer connections are associated with externalizing and internalizing problems and EDs in adolescents [26,27]. Parents' difficulties in identifying their children's emotional and eating problems have also been documented, which is reflected in a discrepancy in the reporting of these problems between adolescents and their parents limiting the seeking of timely treatment for their children [28].
-From a scientific point of view the analyses presented in the text are not reliable, since the research group was so small and consisted of persons with many different eating disorders. As I was pointing out in the previous revision each eating disorder will be different and differently corresponding to externalizing or internalizing behaviors.
Response:
Dear reviewer, we agree that the differences between the diagnostic entities of eating disorders are important, however, due to having such a small sample, it was decided to include them without distinction in the analysis and this was added as a suggestion in the following lines:
Future research should include a review of participants' medical histories to identify comorbidities and incorporate other diagnoses. Similarly, it is suggested that the sample size be increased and that independent analyses be conducted among the diagnostic entities of eating disorders.
-The study you've conducted serves as a good start for very important research and scientific paper. Please expand your research, collect more data from persons with different eating disorders diagnoses, equalize the number of participants from both countries (also taking into consideration approximately equal number of respondents with each ED diagnosis) and then analyze your data again.
Dear reviewer, we agree that one of the important limitations of this study is the small sample size and we would love to recruit more participants for the study. However, since these are clinical samples, this process would take too long. We believe that the results presented in this work are of particular interest, particularly for the Latino population, where further research is needed in the field of eating disorders. Likewise, the sample size was highlighted in different parts of the manuscript as a limitation of the study and its results.
It could also be hypothesized that there was greater recognition of problems in Colombian adolescents compared to Mexican adolescents; although discrepancies in sample size could also have influenced these results, given that the number of participants evaluated in Colombia was more than double that of Mexico
The limitations of this study lie in the small and unequal sample sizes between Mexico and Colombia and the convenience sampling in both countries, making it difficult to generalize the results to either country or even the region. It is important to acknowledge that the small and unequal sample sizes by country (n = 17 for Colombia, n = 8 for Mexico) represent a limitation that may have influenced the statistical power of the study. Small sample sizes make it hard to find significant effects, especially if they are small or moderate. Additionally, group size imbalance can affect the stability of estimates and limit the generalizability of the findings. In this regard, although effect sizes were reported alongside p-values, they should be interpreted with caution, as effect sizes in small samples can be unstable or overestimated [45].
This study provides evidence on the relationship between internalizing and externalizing behaviors in a clinical sample of adolescents from Colombia and Mexico with eating disorders and includes the perspectives of parents and their children. Therefore, it is suggested that the results of this study be interpreted with caution due to the sample size.

Reviewer 3 Report
Comments and Suggestions for Authors
Thank you for addressing the comments and providing additional details to the paper.
A few additional minor comments:
- Table 1: Please add effect sizes and p-values to assess the differences between the two countries.
- Table 2: I apologise if the previous suggestion was not clear. What I meant in my previous comment was to remove the kurtosis, skewness, and Shapiro-Wilk test.
- Figure: typing error: 'Withdrwal'
- Discussion: Please also discuss the significant differences between the sociodemographics of the two countries (Table 1 results), if the authors found any from the analysis.
Author Response
- Table 1: Please add effect sizes and p-values to assess the differences between the two countries.
Response:
The following was added in response to the reviewer's comments
The objective of this study was to characterize the relationship between internalizing and externalizing behaviors in adolescents with EDs from Mexico and Colombia, based on adolescents' self-reports and parental reports. A total of 25 adolescents (2 males and 23 females) aged 12 to 18 years were evaluated (Males: M=13; SD=1.41; Females: M=15.3; SD=1.66). The average age of the Colombian sample was 15.4 years (SD=1.8) and of the Mexican sample 14.1 (SD=2.15), Table 1 shows the description of the sociodemographic characteristics according to the country and the estimation of differences for each country based on sociodemographic variables.
Table 1.
Description of the sociodemographic characteristics by country and the estimation of differences for each country based on the sociodemographic variables.
|
Sociodemographic characteristics |
Frequency (Percentage) |
Comparison analysis |
CBCL |
YSR |
EAT-26 |
|||||||||||||
|
Colombian |
AN-D |
AI-D |
QS |
PI |
RR |
CA |
PE |
AN-D |
AI-D |
QS |
PI |
RR |
CA |
PE |
||||
|
Gender |
Male |
2 (7.7) |
Statistic value |
-.656a |
13.5b |
10 b |
-.47 a |
13 b |
-108 a |
-.914a |
-2.66a |
-1.03a |
-.703 a |
-2.63a |
9 b |
1.0b |
-2.79a |
-.396a |
|
Female |
15 (57.7) |
p |
.618 |
.879 |
.50 |
.714 |
.822 |
.46 |
.506 |
.073 |
.321 |
.595 |
.036 |
.409 |
.043 |
.117 |
.749 |
|
|
ES |
-.546 |
.10 |
.333 |
-.406 |
.133 |
-.91 |
-.725 |
-1.347 |
-.376 |
-.569 |
-1.151 |
.400 |
.933 |
-1.675 |
-.305 |
|||
|
Parents' Gender |
Male |
5 (19.2) |
Statistic value |
-.989a |
22.5b |
27.5b |
-.819a |
22b |
.06a |
-.408a |
-21.9a |
-14.5a |
-19.5a |
-23.1a |
19b |
25.5b |
-10a |
-.379a |
|
Female |
12 (46.2) |
p |
.34 |
.451 |
.832 |
.431 |
.425 |
.953 |
.691 |
.056 |
.203 |
.082 |
.055 |
.265 |
.672 |
.344 |
.713 |
|
|
ES |
-.4667 |
.250 |
.083 |
-.409 |
.267 |
.029 |
-.203 |
-11.21 |
-.83 |
-.998 |
-12.59 |
.367 |
.150 |
-.517 |
-.1903 |
|||
|
Grade |
Sixth |
1 (3.8) |
Statistic value |
NE |
NE |
NE |
NE |
NE |
NE |
NE |
NE |
NE |
NE |
NE |
NE |
NE |
NE |
NE |
|
Seventh |
2 (7.7) |
|||||||||||||||||
|
Eighth |
0 (0) |
p |
NE |
NE |
NE |
NE |
NE |
NE |
NE |
NE |
NE |
NE |
NE |
NE |
NE |
NE |
NE |
|
|
Ninth |
5 (19.2) |
|||||||||||||||||
|
Tenth |
3 (11.5) |
ES |
NE |
NE |
NE |
NE |
NE |
NE |
NE |
NE |
NE |
NE |
NE |
NE |
NE |
NE |
NE |
|
|
Eleventh |
6 (23.1) |
|||||||||||||||||
|
Twelfth |
0 (0) |
|||||||||||||||||
|
Parents' Occupation |
Self-employed |
4 (15.4) |
Statistic value |
NE |
NE |
NE |
NE |
NE |
NE |
NE |
NE |
NE |
NE |
NE |
NE |
NE |
NE |
NE |
|
Employee |
12 (46.2) |
p |
NE |
NE |
NE |
NE |
NE |
NE |
NE |
NE |
NE |
NE |
NE |
NE |
NE |
NE |
NE |
|
|
Homemaker |
0 (0) |
ES |
NE |
NE |
NE |
NE |
NE |
NE |
NE |
NE |
NE |
NE |
NE |
NE |
NE |
NE |
NE |
|
|
Other |
1 (3.8) |
|||||||||||||||||
|
Mexican |
||||||||||||||||||
|
Gender |
Male |
0 (0) |
Statistic value |
NE |
NE |
NE |
NE |
NE |
NE |
NE |
NE |
NE |
NE |
NE |
NE |
NE |
NE |
NE |
|
Female |
9 (34.6) |
p |
NE |
NE |
NE |
NE |
NE |
NE |
NE |
NE |
NE |
NE |
NE |
NE |
NE |
NE |
NE |
|
|
ES |
NE |
NE |
NE |
NE |
NE |
NE |
NE |
NE |
NE |
NE |
NE |
NE |
NE |
NE |
NE |
|||
|
Parents' Gender |
Male |
2 (7.7) |
Statistic value |
-1.43a |
5.50b |
4.50b |
-1.35a |
2.50b |
1.237a |
1.618a |
-.219a |
-2.27a |
.826a |
.107a |
7b |
5b |
-.883a |
4,17a |
|
Female |
7 (26.9) |
p |
.232 |
.767 |
.551 |
.295 |
.219 |
.262 |
.151 |
.854 |
.058 |
.551 |
.930 |
1 |
.659 |
.435 |
.112 |
|
|
ES |
-.945 |
.214 |
.357 |
-.989 |
.643 |
.661 |
.930 |
-.187 |
-1.281 |
.787 |
.099 |
0 |
.286 |
-.588 |
.876 |
|||
|
Grade |
Sixth |
1 (3.8) |
Statistic value |
NE |
NE |
NE |
NE |
NE |
NE |
NE |
NE |
NE |
NE |
NE |
NE |
NE |
NE |
NE |
|
Seventh |
1 (3.8) |
|||||||||||||||||
|
Eighth |
3 (11.5) |
p |
NE |
NE |
NE |
NE |
NE |
NE |
NE |
NE |
NE |
NE |
NE |
NE |
NE |
NE |
NE |
|
|
Ninth |
1 (3.8) |
|||||||||||||||||
|
Tenth |
0 (0) |
ES |
NE |
NE |
NE |
NE |
NE |
NE |
NE |
NE |
NE |
NE |
NE |
NE |
NE |
NE |
NE |
|
|
Eleventh |
2 (7.7) |
|||||||||||||||||
|
Twelfth |
1 (3.8) |
|||||||||||||||||
|
Parents' Occupation |
Self-employed |
2 (7.7) |
Statistic value |
.653c |
4.86d |
4.03d |
1.41c |
.275d |
.146c |
.147c |
7.25c |
3.14c |
1.55c |
10.74c |
1.48d |
0d |
.377c |
1.72c |
|
Employee |
4 (15.4) |
p |
.554 |
.088 |
.133 |
.316 |
.871 |
.867 |
.866 |
.123 |
.116 |
.287 |
.010 |
.477 |
1 |
.701 |
.288 |
|
|
Homemaker |
3 (11.5) |
ES |
.165 |
.6072 |
.5036 |
.562 |
.0344 |
.165 |
.165 |
NE |
-.833 |
-.528 |
1.613 |
.1850 |
0 |
NE |
.222 |
|
Note: AN-D = Anxious Depressed; AI-D = Isolated Depressed; QS = Somatic Complaints; PI = Internalizing Problems; RR = Rule Breaking; CA = Aggressive Behavior; PE = Externalizing Problems."; ES= Effect Size. a: The comparison statistic used was Welch's t-test, and the effect size measure was Cohen's d. b: The comparison statistic used was Mann-Whitney U, and the effect size measure was Biserial correlation by ranks. c= The comparison statistic used was ANOVA and the effect size was omega squared (ω2). d= The comparison statistic used was Kruskal Wallis and the effect size was epsilon squared (ε2). NE= Not estimable due to insufficient observations.
When comparing the variables of interest according to sociodemographic characteristics in each country, it was observed that, in the Colombian sample, there were statistically significant differences between sexes on the YSR internalizing problems variable, with higher scores among women (M = 27.87; SD = 10.52) than among men (M = 19; SD = 2.83). Similarly, on the YSR aggressive behavior variable, women also obtained higher scores (M = 12.67; SD = 4.03) compared to men (M = 5; SD = 2.83).
In the Mexican sample, statistically significant differences were found according to the parents' occupation in the YSR internalizing problems variable, between the employee (M = 29.75; SD = 4.99) and independent workers (M = 16; SD = 1.41) levels, as well as between employees and homemakers (M = 19.33; SD = 2.52); however, no significant differences were observed between independent workers and homemakers.
- Table 2: I apologise if the previous suggestion was not clear. What I meant in my previous comment was to remove the kurtosis, skewness, and Shapiro-Wilk test.
Response
We don't understand clearly. Is the suggestion to eliminate these indicators of normalcy?
- Figure: typing error: 'Withdrwal'
Response:
The error was corrected
- Discussion: Please also discuss the significant differences between the sociodemographics of the two countries (Table 1 results), if the authors found any from the analysis.
Response
Dear reviewer, these analyses detected very few significant differences in this analysis and were of little interest. In addition to the limited space allotted by the journal in the manuscript, it was decided not to include these results in the discussion.
